# VATT: Transformers for Multimodal Self-Supervised Learning from Raw Video, Audio and Text

**Hassan Akbari**[*]
Columbia University
ha2436@columbia.edu

**Liangzhe Yuan**
Google
lzyuan@google.com

**Rui Qian**[*]
Cornell University
rq49@cornell.edu

**Wei-Hong Chuang**
Google
whchuang@google.com

**Shih-Fu Chang**
Columbia University
sc250@columbia.edu

**Yin Cui**
Google
yincui@google.com

**Boqing Gong**
Google
bgong@google.com

## Abstract

We present a framework for learning multimodal representations from unlabeled data using convolution-free Transformer architectures. Specifically, our **V**ideo-**A**udio-**T**ext **T**ransformer (**VATT**) takes raw signals as inputs and extracts multimodal representations that are rich enough to benefit a variety of downstream tasks. We train VATT end-to-end from scratch using multimodal contrastive losses and evaluate its performance by the downstream tasks of video action recognition, audio event classification, image classification, and text-to-video retrieval. Furthermore, we study a modality-agnostic, single-backbone Transformer by sharing weights among the three modalities. We show that the convolution-free VATT outperforms state-of-the-art ConvNet-based architectures in the downstream tasks. Especially, VATT's vision Transformer achieves the top-1 accuracy of 82.1% on Kinetics-400, 83.6% on Kinetics-600, 72.7% on Kinetics-700, and 41.1% on Moments in Time, new records while avoiding supervised pre-training. Transferring to image classification leads to 78.7% top-1 accuracy on ImageNet compared to 64.7% by training the same Transformer from scratch, showing the generalizability of our model despite the domain gap between videos and images. VATT's audio Transformer also sets a new record on waveform-based audio event recognition by achieving the mAP of 39.4% on AudioSet without any supervised pre-training. VATT's source code is publicly available.[2]

## 1 Introduction

Convolutional neural networks (CNNs) [53, 51] have triumphed over various computer vision tasks. The inductive bias induced by convolutions, namely translation invariance and locality, are proven effective for the visual data. In the meantime, however, we witness in the natural language processing (NLP) community a paradigm shift from the models with strong inductive biases, such as recurrent neural networks [43, 7] and CNNs [104, 32], to more general architectures constructed upon self-attention. Particularly, Transformers [88] have become the de facto model architecture for NLP

---

[*]Work done during an internship at Google.

[2]https://github.com/google-research/google-research/tree/master/vatt

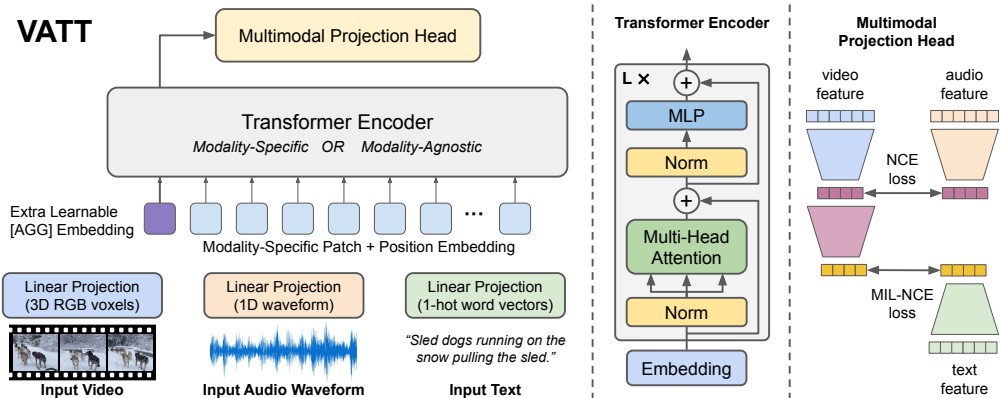

Figure 1: **Overview of the VATT architecture and the self-supervised, multimodal learning strategy**. VATT linearly projects each modality into a feature vector and feeds it into a Transformer encoder. We define a semantically hierarchical common space to account for the granularity of different modalities and employ the Noise Contrastive Estimation (NCE) to train the model.

tasks [23, 70, 71, 10]. Pre-training a Transformer on large text corpora followed by fine-tuning gives rise to state-of-the-art results for different downstream tasks.

In view of the success of the attention mechanism in NLP, there has been a rich line of works exploring its potential in computer vision. Early work studied hybrid models consisting of both convolutions and attention modules [89, 94, 36, 105]. Recent studies showed that convolution-free, specially designed all-attention models can match CNNs' performance on image recognition tasks [106, 44, 73]. Most recently, [25] achieved impressive performance on several image recognition tasks, including ImageNet [22], using a pre-trained Transformer with minimal architecture changes. Their work delivered a compelling message that "large scale (supervised) training trumps inductive bias (for image classification)." This conclusion was further extended to video recognition tasks by [9, 5].

However, the large-scale supervised training of Transformers is essentially troubling for two main reasons. First, it rules out the much larger other part of "big visual data," i.e, the vast amount of unlabeled, unstructured visual data. As a result, the supervised training strategy could produce biased systems that require even more labeled data to correct their biases. Second, this strategy fundamentally limits the application scope of Transformers in computer vision because it is costly and extremely time-consuming to collect enough labeled images or videos for training the millions of parameters, choosing hyper-parameters, and validating their expected generalization.

Hence, this work poses another pressing question about the Transformers that take raw signals as input. *How to empower them with large-scale, unlabeled visual data?* To answer this question, we draw insights from NLP. BERT [23] and GPT [70, 71, 10] use masked language modeling as their pre-training tasks. Natural languages are organic supervision for Transformers. They sequentially place words, phrases, and sentences into context, granting them semantics and syntax. For visual data, *the most organic supervision is arguably the multimodal videos.* They are abundantly available in the digital world, and their temporal, cross-modality regulation, and therefore supervision, requires no human annotation. The extreme scale of multimodal videos is potentially capable to teach Transformers necessary priors, as opposed to predefined inductive biases, to model the visual world.

To this end, we study self-supervised, multimodal pre-training of three Transformers [88], which take as input the raw RGB frames of internet videos, audio waveforms, and text transcripts of the speech audio, respectively. We call the video, audio, text Transformers VATT. Figure 1 illustrates the architecture. VATT borrows the exact architecture from BERT [23] and ViT [25] except the layer of tokenization and linear projection reserved for each modality separately. This design shares the same spirit as ViT that we make the minimal changes to the architecture so that the learned model can transfer its weights to various frameworks and tasks. Furthermore, the self-supervised, multimodal learning strategy resonates the spirit of BERT and GPT that the pre-training requires minimal human curated labels.

We evaluate the pre-trained Transformers on a variety of downstream tasks: *image classification, video action recognition, audio event classification, and zero-shot text-to-video retrieval*. Fine-tuning

the vision-modality Transformer on ImageNet [22] obtains the top-1 accuracy of 78.7%, which is comparable to 79.9% achieved by ViT. This result is especially appealing considering the domain gap between videos and images, and that ViT is pre-trained using a large-scale, human-curated image dataset. Furthermore, we set new records on Kinetics-400 [14], Kinetics-600 [15], Moments in Time [61], and AudioSet [33] without supervised pre-training.

Our VATT results, along with others reported for NLP tasks [23, 10], image recognition [25], semantic segmentation [108], point cloud classification [107], and action recoginition [9], demonstrate that Transformer is a versatile general-purpose architecture for different types of data.

To move one step forward, we challenge the Transformers in VATT by a seemingly too strong constraint: sharing weights among the video, audio, and text modalities. The idea is to test whether there exists a single, general-purpose model for all the modalities — of course, they still have their own layers of tokenization and linear projection. Preliminary results are encouraging. This modality-agnostic Transformer is on par with three modality-specific ones of slightly smaller sizes.

Finally, another contribution of this work is DropToken, a simple and yet effective technique to reduce the training complexity with a minor reduction of the end Transformers' performance. DropToken randomly drops a portion of the video and audio tokens from each input sequence during training, allowing for high-resolution inputs and leveraging their abundance. This is significant for Transformers because their computational complexity is quadratic with respect to the number of input tokens.

## 2    Related work

### 2.1    Transformers in Vision

Transformer was originally built for NLP tasks [88] and the design of multi-head attention shows its effectiveness on modeling long-term correlation of words. A few attempts have been made to use Transformer for vision tasks like image super-resolution [99], object detection [11] and multimodal video understanding [84, 19, 57]. However these methods still rely on the feature extracted by CNNs. Recently, [25] proposes a set of convolution-free vision Transformers which directly work on raw images and obtain competitive performance with CNNs. [86] improves the training data efficiency of [25] by using stronger data augmentations and knowledge distillation. Since then, the pure Transformer design has been adopted to various vision tasks including semantic segmentation [108], point cloud classification [107], action recoginition [9, 78, 5]. To the best of our knowledge, our VATT is the first Transformer model on raw multimodal inputs of video, audio and text.

### 2.2    Self-Supervised Learning

**Single vision modality.** Early work of self-supervised visual representation learning usually learns from unlabeled images via manually specified pretext tasks, like auto-encoding [64, 102, 103], patch location prediction [24], solving jigsaw puzzles [63], and image rotation prediction [35]. [95] propose a novel instance discrimination objective. The recent trend of contrastive learning [40, 17, 100, 37, 41, 85] integrates data augmentations and instance discrimination by maintaining relative consistency between representations of an image and its augmented view. Clustering can also provide an effective addition [12]. Recently, [18] conduct contrastive learning using ViT [25] and achieve impressive results. As for the video domain, it is natural to exploit the temporal signals as the pretext task. Examples include predicting the future frame [82], motion and appearance statistics [90], speed [8, 91] and encodings [56, 38, 39], sorting frames or video clips [54, 97, 45, 31]. Recently, [68] apply contrastive learning to videos with a temporal sampling strategy and temporally consistent spatial augmentation.

**Multimodal video.** Video is a natural source of multimodal data. Multimodal self-supervised learning can be achieved by predicting whether a video has correspondence with an audio stream [3, 4, 62, 50], cross-modality clustering [2], and evolving losses [67]. Recently, [1] use contrastive loss to learn from video, audio and text; [74] learn to predict a broad view that spans a longer temporal context from a narrow view. VATT serves as a first work combining the strength of convolution-free Transformer and multimodal contrastive learning.

# 3 Approach

In this section, we introduce our convolution-free VATT architecture and elaborate on the self-supervised multimodal objectives for training VATT from scratch.

Figure 1 is an overview of the architecture. We feed each modality to a tokenization layer, where the raw input is projected to an embedding vector followed by a Transformer. There are two major settings: 1) The backbone Transformers are separate and have specific weights for each modality, and 2) The Transformers share weights, namely, there is a single backbone Transformer applied to any of the modalities. In either setting, the backbone extracts modality-specific representations, which are then mapped to common spaces to be compared with each other by contrastive losses. We describe each module in the following.

## 3.1 Tokenization and Positional Encoding

VATT operates on raw signals. The vision-modality input consists of 3-channel RGB pixels of video frames, the audio input is in the form of air density amplitudes (waveforms), and the text input is a sequence of words. We first define a modality-specific tokenization layer that takes as input the raw signals and returns a sequence of vectors to be fed to the Transformers. Besides, each modality has its own positional encoding, which injects the order of tokens into Transformers [88]. We partition an entire video clip of size $T \times H \times W$ to a sequence of $\lceil T/t \rceil \cdot \lceil H/h \rceil \cdot \lceil W/w \rceil$ patches, where each patch contains $t \times h \times w \times 3$ voxels. We apply a linear projection on the entire voxels in each patch to get a $d$-dimensional vector representation. This projection is performed by a learnable weight $\boldsymbol{W}_{vp} \in \mathbb{R}^{t \cdot h \cdot w \cdot 3 \times d}$. This can be seen as a 3D extension of the patching mechanism proposed in [25]. To encode the position of these patches, we define a dimension-specific sequence of learnable embeddings as follows:

$$\boldsymbol{e}_{i,j,k} = \boldsymbol{e}_{\text{Temporal}_i} + \boldsymbol{e}_{\text{Horizontal}_j} + \boldsymbol{e}_{\text{Vertical}_k},$$
$$\boldsymbol{E}_{\text{Temporal}} \in \mathbb{R}^{\lceil T/t \rceil \times d}, \quad \boldsymbol{E}_{\text{Horizontal}} \in \mathbb{R}^{\lceil H/h \rceil \times d}, \quad \boldsymbol{E}_{\text{Vertical}} \in \mathbb{R}^{\lceil W/w \rceil \times d} \tag{1}$$

where $\boldsymbol{e}_i$ is the $i$-th row of $\boldsymbol{E}$. This scheme allows us to use $\lceil T/t \rceil + \lceil H/h \rceil + \lceil W/w \rceil$ positional embeddings to encode all the $\lceil T/t \rceil \cdot \lceil H/h \rceil \cdot \lceil W/w \rceil$ patches in a video clip. The raw audio waveform is a 1D input with length $T'$, and we partition it to $\lceil T'/t' \rceil$ segments each containing $t'$ waveform amplitudes. Similar to video, we apply a linear projection with a learnable weight $\boldsymbol{W}_{ap} \in \mathbb{R}^{t' \times d}$ to all elements in a patch to get a $d$-dimensional vector representation. We use $\lceil T'/t' \rceil$ learnable embeddings to encode the position of each waveform segment. For text, we first construct a vocabulary of size $v$ out of all words in our training dataset. For an input text sequence, we then map each word to a $v$-dimensional one-hot vector followed by a linear projection with a learnable weight $\boldsymbol{W}_{tp} \in \mathbb{R}^{v \times d}$. This is equivalent to an embedding dictionary lookup, which has been widely used in natural language understanding [60].

### 3.1.1 DropToken

We introduce DropToken, a simple and yet effective strategy to reduce the computational complexity during training. Once we get the token sequence for the video or audio modality, we randomly sample a portion of the tokens and then feed the sampled sequence, not the complete set of tokens, to the Transformer. This is crucial for reducing the computational cost because a Transformer's computation complexity is quadratic, $O(N^2)$, where $N$ is number of tokens in the input sequence. Any effort on reducing the input length would reduce the number of FLOPs quadratically. This has an immediate impact on the wall clock time for training these models and makes it possible to host large models in limited hardware. We argue that instead of reducing the resolution or dimension of the raw inputs, it is better to keep a high-fidelity input and randomly sample the tokens via DropToken. DropToken is appealing especially with the raw video and audio inputs, which may contain high redundancies.

## 3.2 The Transformer Architecture

For simplicity, we adopt the most established Transformer architecture [23], which has been widely used in NLP. Similar to ViT [25], we do not tweak the architecture so that our weights can be easily transferred to any standard Transformer implementation. We will briefly elaborate on the pipeline (also illustrated in Figure 1 middle panel) and refer the reader to [25, 23] for more details of the

standard Transformer architecture. The sequence of input tokens to the Transformer follows the below formulation:

$$\boldsymbol{z}_{\text{in}} = [\boldsymbol{x}_{\text{AGG}};\ \boldsymbol{x}_0 \boldsymbol{W}_P;\ \boldsymbol{x}_1 \boldsymbol{W}_P; \ldots;\ \boldsymbol{x}_N \boldsymbol{W}_P] + \boldsymbol{e}_{\text{POS}} \tag{2}$$

where $\boldsymbol{x}_n$ is the input patches sequence and $\boldsymbol{x}_{\text{AGG}}$ is the learnable embedding of a special aggregation token whose corresponding output in the Transformer ($z_{\text{out}}^0$) is used as the aggregated representation for the entire input sequence. This will be later used for classification and common space mapping. We use a standard self-attention [88] as the Multi-Head-Attention (MHA) module, and GeLU [42] as the activation in the MLP layer. We also use Layer Normalization [6] before the MHA and MLP modules. In our text model, we remove the position encoding $\boldsymbol{e}_{\text{POS}}$ and add a learnable relative bias to each attention score of the first layer in the MHA module. This simple change makes our text model's weights directly transferable to the state-of-the-art text model T5 [72].

### 3.3 Common Space Projection

We use common space projection and contrastive learning in that common space to train our networks. More specifically, given a video-audio-text triplet, we define a semantically hierarchical common space mapping that enables us to directly compare video-audio pairs as well as video-text pairs by the cosine similarity. As argued in [1], such comparison is more feasible if we assume there are different levels of semantic granularity for these modalities. To achieve this, we define multi-level projections as follows:

$$\begin{aligned}
\boldsymbol{z}_{v,va} &= g_{v\to va}(\boldsymbol{z}_{\text{out}}^{\text{video}}), & \boldsymbol{z}_{a,va} &= g_{a\to va}(\boldsymbol{z}_{\text{out}}^{\text{audio}}) \\
\boldsymbol{z}_{t,vt} &= g_{t\to vt}(\boldsymbol{z}_{\text{out}}^{\text{text}}), & \boldsymbol{z}_{v,vt} &= g_{v\to vt}(\boldsymbol{z}_{v,va})
\end{aligned} \tag{3}$$

where $g_{v\to va}$ and $g_{a\to va}$ are the projection heads to respectively map the video and audio Transformers' outputs to the video-audio common space $\mathcal{S}_{va}$. Moreover, $g_{t\to vt}$ and $g_{v\to vt}$ project the text Transformer's outputs and the video embedding in the $\mathcal{S}_{va}$ space to video-text common space, $\mathcal{S}_{vt}$. This multi-level common space projection is depicted in Figure 1 (the rightmost panel). The main intuition behind this hierarchy is that different modalities have different levels of semantic granularity, so we should impose this as an inductive bias in the common space projection. Similar to [1], we use a linear projection for $g_{a\to va}(.)$, $g_{t\to vt}(.)$, and $g_{v\to vt}(.)$, and a two-layer projection with ReLU in between for $g_{v\to va}(.)$. To ease the training, a batch normalization is used after each linear layer.

### 3.4 Multimodal Contrastive Learning

Inspired by [1, 3, 59], we use Noise Contrastive Estimation (NCE) to align video-audio pairs and Multiple Instance Learning NCE (MIL-NCE) to align video-text pairs. The pairs are composed from different temporal locations in the video-audio-text stream. Positive pairs from two modalities are constructed by sampling their corresponding streams from the same location in the video, and negative pairs are constructed by sampling from any non-matching locations in the video [1]. Concretely, given the common space specified in Section 3, the loss objectives can be written as follows:

$$\text{NCE}(\boldsymbol{z}_{v,va}, \boldsymbol{z}_{a,va}) = -\log\left(\frac{\exp(\boldsymbol{z}_{v,va}^\top \boldsymbol{z}_{a,va}/\tau)}{\exp(\boldsymbol{z}_{v,va}^\top \boldsymbol{z}_{a,va}/\tau) + \sum_{z'\in\mathcal{N}}\exp(\boldsymbol{z'}_{v,va}^\top \boldsymbol{z'}_{a,va}/\tau)}\right), \tag{4}$$

$$\text{MIL-NCE}(\boldsymbol{z}_{v,vt}, \{\boldsymbol{z}_{t,vt}\}) = -\log\left(\frac{\sum_{\boldsymbol{z}_{t,vt}\in\mathcal{P}}\exp(\boldsymbol{z}_{v,vt}^\top \boldsymbol{z}_{t,vt}/\tau)}{\sum_{\boldsymbol{z}_{t,vt}\in\mathcal{P}}\exp(\boldsymbol{z}_{v,vt}^\top \boldsymbol{z}_{t,vt}/\tau) + \sum_{z'\in\mathcal{N}}\exp(\boldsymbol{z'}_{v,vt}^\top \boldsymbol{z'}_{t,vt}/\tau)}\right), \tag{5}$$

where $\mathcal{N}$ contains all non-matching pairs in a batch. In Equation 5, $\mathcal{P}$ contains five text clips that are nearest neighbors to the video clip in time. $\tau$ is a temperature to adjust the softness of the objectives in distinguishing the positive pairs from the negative pairs.

The overall per-sample objective for training the entire VATT model end-to-end is as follows:

$$\mathcal{L} = \text{NCE}(\boldsymbol{z}_{v,va}, \boldsymbol{z}_{a,va}) + \lambda\text{MIL-NCE}(\boldsymbol{z}_{v,vt}, \{\boldsymbol{z}_{t,vt}\}), \tag{6}$$

where $\lambda$ balances the two losses. The model is optimized based on the back-propagation of the average loss calculated over a batch of samples.

# 4 Experiments

In this section, we first briefly describe the experimental setup for the pre-training and downstream evaluation, and then present the results and analytic interpretation of VATT in different tasks. We refer the reader to the Appendix for a more detailed description of all experimental settings.

## 4.1 Experimental Setup

**Pre-train:** we use a combination of AudioSet [33] and HowTo100M [58] datasets to pre-train VATT— we use only a subset of the HowTo100M dataset in compliance with Youtube's policies. Following [1], we use video-audio-text triplets from HowTo100M clips while only using video-audio pairs from AudioSet. We sample 32 frames at 10 fps with a spatial size of $224 \times 224$ following a random crop, horizontal flip and color augmentation (details in A.2.1). Accordingly, we sample audio waveforms in sync at 48kHz. Both video and audio are normalized between [-1,1]. We use patch sizes of $4 \times 16 \times 16$ and 128 for video and raw waveform tokenization, respectively (ablation in A.5). We use one-hot vectors to encode text sequences (capped to 16 tokens) with the vocabulary size of $2^{16}$. In all pre-training experiments, we use DropToken with drop rate 50%. We train our models using the Adam optimizer [46] with a quarter-period cosine scheduled learning rate from $1e$-4 to $5e$-5 and 10k warmup steps. Optimization is performed on totally 500k steps with batch size 2048 (512 in exploration experiments). Following the previously established practice [1] for the projection to the common spaces $\mathcal{S}_{va}$ and $\mathcal{S}_{vt}$, we use $d_{va} = 512$ and $d_{vt} = 256$. We also use the temperature of $\tau = 0.07$ and the weight of $\lambda = 1$ in the loss in Equation 6. We use 4 network sizes in our experiments (details in A.2.2). We use the Medium model (155M parameters) for our modality-agnostic variant (VATT-MA), and 3 variants for the modality-specific video-audio-text backbones: Base-Base-Small (BBS; 197M), Medium-Base-Small (MBS; 264M), and Large-Base-Small (LBS; 415M). Pre-training an MBS VATT with batch size 2048 on 256 TPUs (v3) takes less than 3 days. Pre-training with batch size 512 takes less than 1 day.

**Downstream:** we evaluate the pre-trained VATT models on 4 major downstream tasks using a total of 10 datasets. We use UCF101 [81], HMDB51 [52], Kinetics-400 [14], Kinetics-600 [15], and Moments in Time [61] for video action recognition. We use ESC50 [66] and AudioSet [33] for audio event classification, and we evaluate the quality of our video-text common space representations by zero-shot text-to-video retrieval on YouCook2 [109] and MSR-VTT [98]. Finally, we evaluate the transferability of the vision backbone by fine-tuning it on ImageNet classification [22]. Since HMDB51, UCF101, and ESC50 are very small datasets compared to the size of our networks, we only use them to train a linear classifier on top of the frozen pre-trained backbones. In our exploration experiments, we report linear classification accuracy and zero-shot video retrieval metrics. We refer to the Appendix for a detailed description of the datasets and the experimental setup.

## 4.2 Results

### 4.2.1 Fine-tuning for video action recognition

We fine-tune VATT's vision Transformer on Kinetics-400, Kinetics-600, and Moments in Time, three of the arguably most established large-scale datasets for video action recognition. We use the final checkpoints of four pre-train settings for these experiments: three modality-specific variations (*LBS, MBS, BBS*), and one modality-agnostic (*Medium*). Table 1 shows the results compared with the state-of-the-art video models. On all three datasets, we achieve higher accuracy than previous works including TimeSFormer [9], a recent effort in fine-tuning the ViT checkpoints obtained by *supervised* pre-training. In contrast, our pre-training does not rely on any labels curated by humans. To the best of our knowledge, VATT provides the first vision Transformer backbone that is pre-trained from scratch using self-supervision on multimodal videos and achieves state-of-the-art results on video action recognition. It is also worth mentioning that fine-tuning VATT on the most recent Kinetics-700 dataset results in a top-1 accuracy of 72.7%, which outperforms the state-of-the-art top-1 accuracy of 72.4% in [47].

To further quantify how much the multimodal self-supervised pre-training helps in achieving these numbers, we train a variant from scratch without any pre-training and observe the top-1 and top-5 accuracies of 26.4% and 51.8% on Kinetics-400, respectively. The low accuracies verify the efficacy of our pre-training strategy for VATT. Finally, we find that VATT-MA-Medium, the modality-agnostic

| Method | Kinetics-400 Top-1 | Kinetics-400 Top-5 | Kinetics-600 Top-1 | Kinetics-600 Top-5 | Moments in Time Top-1 | Moments in Time Top-5 | TFLOPs |
|---|---|---|---|---|---|---|---|
| I3D [13] | 71.1 | 89.3 | 71.9 | 90.1 | 29.5 | 56.1 | - |
| R(2+1)D [26] | 72.0 | 90.0 | - | - | - | - | 17.5 |
| bLVNet [27] | 73.5 | 91.2 | - | - | 31.4 | 59.3 | 0.84 |
| S3D-G [96] | 74.7 | 93.4 | - | - | - | - | - |
| Oct-I3D+NL [20] | 75.7 | - | 76.0 | - | - | - | 0.84 |
| D3D [83] | 75.9 | - | 77.9 | - | - | - | - |
| I3D+NL [93] | 77.7 | 93.3 | - | - | - | - | 10.8 |
| ip-CSN-152 [87] | 77.8 | 92.8 | - | - | - | - | 3.3 |
| AttentionNAS [92] | - | - | 79.8 | 94.4 | 32.5 | 60.3 | 1.0 |
| AssembleNet-101 [77] | - | - | - | - | 34.3 | 62.7 | - |
| MoViNet-A5 [47] | 78.2 | - | 82.7 | - | 39.1 | - | 0.29 |
| LGD-3D-101 [69] | 79.4 | 94.4 | 81.5 | 95.6 | - | - | - |
| SlowFast-R101-NL [30] | 79.8 | 93.9 | 81.8 | 95.1 | - | - | 7.0 |
| X3D-XL [29] | 79.1 | 93.9 | 81.9 | 95.5 | - | - | 1.5 |
| X3D-XXL [29] | 80.4 | 94.6 | - | - | - | - | 5.8 |
| TimeSFormer-L [9] | 80.7 | 94.7 | 82.2 | 95.6 | - | - | 7.14 |
| VATT-Base | 79.6 | 94.9 | 80.5 | 95.5 | 38.7 | 67.5 | 9.09 |
| VATT-Medium | 81.1 | **95.6** | 82.4 | 96.1 | 39.5 | **68.2** | 15.02 |
| VATT-Large | **82.1** | 95.5 | **83.6** | **96.6** | **41.1** | 67.7 | 29.80 |
| VATT-MA-Medium | 79.9 | 94.9 | 80.8 | 95.5 | 37.8 | 65.9 | 15.02 |

Table 1: Video action recognition accuracy on Kinetics-400, Kinetics-600, and Moments in Time.

backbone shared by the video, audio, and text modalities, is on par with the modality-specific VATT-Base when fine-tuned for the video action recognition. This result is encouraging as it indicates the potential of unifying three data modalities by a *single* Transformer backbone.

### 4.2.2 Fine-tuning for audio event classification

We fine-tune VATT's audio Transformer on AudioSet, which benchmarks the task of multi-label audio event classification. We use the final checkpoints of two pre-train settings: one modality-specific (BBS), and one modality-agnostic (Medium). Table 2 shows the results compared to state-of-the-art models. Following common practice [34, 48], we report mean Average Precision (mAP), Area Under Curve (AUC), and d-prime (based on AUC) [34]. Our audio Transformer consistently outperforms the existing CNN-based models in all metrics. More interestingly, fine-tuning the modality-agnostic backbone (VATT-MA-Medium) is on par with fine-tuning the modality-specific one (VATT-Base). To the best of our knowledge, VATT is the first Transformer that outperforms CNN-based models in audio event recognition. VATT operates on raw waveforms and does not utilize any handcrafted features.

### 4.2.3 Fine-tuning for image classification

In this section, we show that our pipeline is capable of transferring the learned knowledge into another domain by performing the image classification task, even though the models are pre-trained in the multimodal video domain. We fine-tune the vision Transformer in VATT-BBS on ImageNet without any modification to the backbone architecture. Instead, to satisfy the voxel-to-patch layer we replicate the input image 4 times and feed it to the network. The network sees the input as a single-frame video clip and performs spatial self-attention. Table 3 shows the results for fine-tuning the vision Transformer end-to-end on ImageNet. We can see that our pre-training leads to a significant boost in the accuracy compared to training from scratch. We also observe that even though the self-supervised pre-training happens in the video domain, we still achieve competitive results to the *supervised* pre-training using large-scale *image* data [25].

### 4.2.4 Zero-shot text-to-video retrieval

We feed video-text pairs to VATT-MBS, and extract representations in the $S_{vt}$ space. We then calculate the similarity between each video-text pair from YouCook2 and MSR-VTT. Given a text query, we rank the videos based on their similarities to the text. We then measure the recall for the

| METHOD | mAP | AUC | d-prime |
|---|---|---|---|
| DaiNet [21] | 29.5 | 95.8 | 2.437 |
| LeeNet11 [55] | 26.6 | 95.3 | 2.371 |
| LeeNet24 [55] | 33.6 | 96.3 | 2.525 |
| Res1dNet31 [49] | 36.5 | 95.8 | 2.444 |
| Res1dNet51 [49] | 35.5 | 94.8 | 2.295 |
| Wavegram-CNN [49] | 38.9 | 96.8 | 2.612 |
| VATT-Base | **39.4** | **97.1** | **2.895** |
| VATT-MA-Medium | 39.3 | 97.0 | 2.884 |

Table 2: Finetuning results for AudioSet event classification.

| METHOD | PRE-TRAINING DATA | TOP-1 | TOP-5 |
|---|---|---|---|
| iGPT-L [16] | ImageNet | 72.6 | - |
| ViT-Base [25] | JFT | **79.9** | - |
| VATT-Base | - | 64.7 | 83.9 |
| VATT-Base | HowTo100M | 78.7 | 93.9 |

Table 3: Finetuning results for ImageNet classification.

| METHOD | BATCH | EPOCH | YouCook2 R@10 | YouCook2 MedR | MSR-VTT R@10 | MSR-VTT MedR |
|---|---|---|---|---|---|---|
| MIL-NCE [59] | 8192 | 27 | **51.2** | **10** | **32.4** | **30** |
| MMV [1] | 4096 | 8 | 45.4 | 13 | 31.1 | 38 |
| VATT-MBS | 2048 | 4 | 45.5 | 13 | 29.7 | 49 |
| VATT-MA-Medium | 2048 | 4 | 40.6 | 17 | 23.6 | 67 |

Table 4: Zero-shot text-to-video retrieval.

correct video in the top-10 videos. We also measure the median of the rank of the correct video. Table 4 compares our video retrieval results to two baselines. In our experiments we observe that the zero-shot retrieval results are heavily affected by the batch size and number of epochs, confirming the observation made in [1]. That said, our model still delivers comparable results to MMV [1] while being pre-trained with a half number of epochs and a half batch size of theirs. We also experiment with a larger batch size 8192 and longer pre-training for 6 epochs, arriving at exactly the same results as MIL-NCE [59] on YouCook2 and the R@10 of 29.2 and MedR of 42 on MSR-VTT. We also notice that, probably due to the noisy nature of text transcripts, a sophisticated language model like ours is underrated. As shown in [1], using a simple linear projection would still perform reasonably well. It is worth exploring other, higher-quality text sources in future work.

#### 4.2.5 Feature visualization

We take our modality-specific and modality-agnostic VATT fine-tuned on Kinetics-400 and visualize their output feature representations using t-SNE. For comparison, we also include the feature visualization of the vision Transformer trained from scratch on Kinetics-400. From Figure 2, we observe that the fine-tuned VATT yields a much better separation than the model trained from scratch. Furthermore, it is worth noting that there is no clear difference between the modality-agnostic features and the modality-specific ones.

We further investigate the VATT backbones without any fine-tuning. We randomly choose 1k video clips from the YouCook2 dataset and store the representations from two points of a pre-trained VATT model. One is after the tokenization layer (input space of the Transformer), and the other is after the common space projection (output space), where the loss is computed. Figure 3-top visualizes the representations, comparing modality-specific VATT to modality-agnostic VATT. Interestingly, we observe that the representations are slightly more mixed together in the modality-agnostic setting compared to the modality-specific ones, implying that the modality-agnostic backbone sees different modalities as different symbols describing the same concept. This is analogous to a unified language model in NLP that supports multiple languages.

To see how well VATT distinguishes positive video-text pairs from randomly sampled pairs, we calculate pair-wise similarities for all possible pairs and perform a Kernel Density Estimation (KDE) to visualize the distributions of the similarities of the positive pairs vs. negative pairs. We perform this procedure for both input and output spaces of the modality-specific and modality-agnostic backbones. Figure 3-bottom shows the KDE curves of these similarities. We can see that VATT in both settings separates the positive and negative pairs in its output space. This verifies VATT's efficacy in learning a semantic common space for different modalities, even if we share the backbone across modalities.

#### 4.2.6 Model Activations

We measure the average activation of the modality-agnostic VATT when a full multimodal input is fed to the model. More specifically, we sample 100k short video clips from the test split of HowTo100M along with their corresponding audio and text and feed them to the model separately. For each

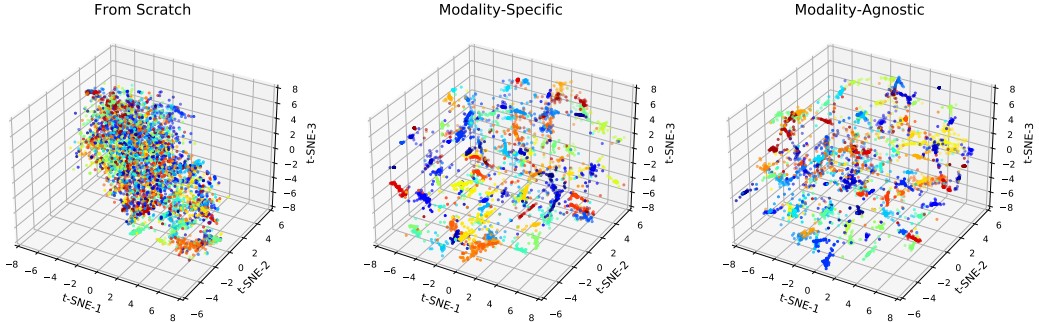

Figure 2: t-SNE visualization of the feature representations extracted by the vision Transformer in different training settings. For better visualization, we show 100 random classes from Kinetics-400.

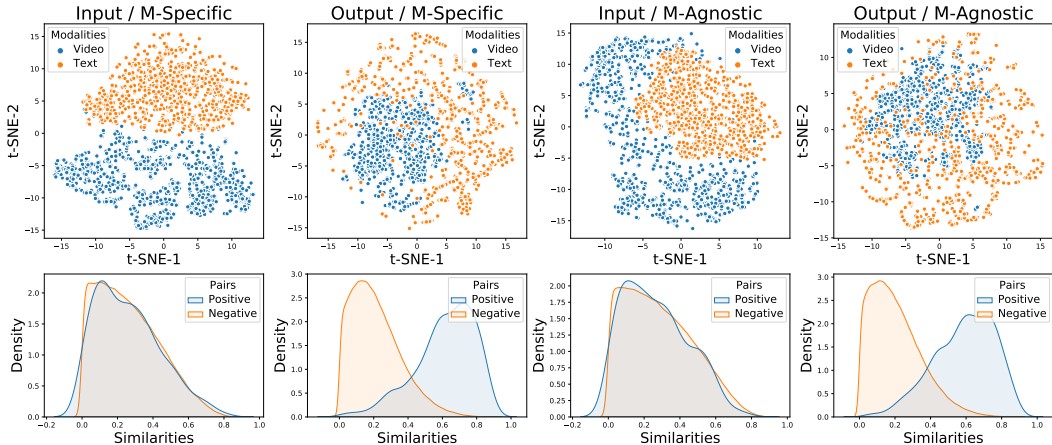

Figure 3: t-SNE visualization and distribution of pair-wise similarities of the input space vs. output space for modality-specific and modality-agnostic backbones when different modalities are fed.

modality, we calculate the average activation of each node at the output of the MLP module, before the residual addition (Figure 1-Transformer Encoder). Figure 4 shows the average activations across all nodes in a Medium-size model. We observe that earlier nodes in the model are activated with the text inputs, while the middle-to-later nodes are activated with video and audio modalities. However, the nodes in the last layers of the network are activated with all modalities almost equally. This might suggest that the model allocates different nodes to certain modalities while reaching the same level of semantic perception for all modalities in the later layers. Such observation encourages further studies on the possibility of utilizing Mixture-of-Experts [79, 28, 76] to increase the model's capacity for simultaneous multimodal perception. We leave this direction of research for future work.

### 4.2.7 Effect of DropToken

We introduced a new method to reduce the redundancy in high-resolution data. To study the effect of the proposed DropToken method on downstream applications and the pre-training computation, we perform pre-training by randomly dropping $75\%$, $50\%$, $25\%$, and $0\%$ (no drop) of the tokens from the video and audio inputs. Table 5 shows the accuracy of linear classification on HMDB51, UCF101, ESC50 and R@10 on YouCook2 and MSR-VTT vs. the drop rate along with GFLOPs during a forward call. We choose $50\%$ sampling rate for our large-scale pre-training as it offers a good trade-off between accuracy and computational costs. We then take the final checkpoint of the pre-trained VATT with $50\%$ DropToken rate and perform fine-tuning on Kinetics-400 at different DropToken rates and at different spatial and temporal resolutions to see how high-resolution inputs coupled with DropToken compare to low-resolution inputs with no tokens dropped during fine-tuning. Table 6 shows the top-1 accuracy on Kinetics-400. We argue against using low-resolution inputs, which is the most common approach to reduce the computational cost during training. Instead, we suggest using high-resolution inputs with DropToken, whose accuracy and training cost are comparable to or better than low-resolution counterparts.

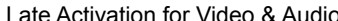

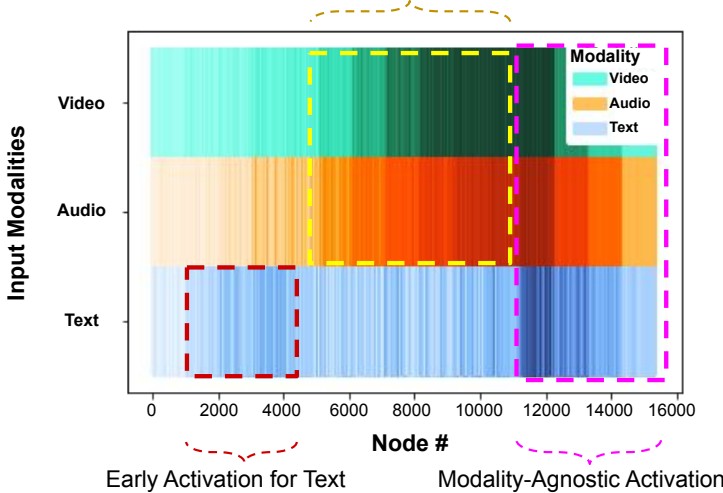

Figure 4: The average node activation across the Modality-Agnostic-Medium VATT while feeding a multimodal video-audio-text triplet to the model.

|  | DropToken Drop Rate | | | |
| --- | --- | --- | --- | --- |
|  | 75% | 50% | 25% | 0% |
| Multimodal GFLOPs | 188.1 | 375.4 | 574.2 | 784.8 |
| HMDB51 | 62.5 | 64.8 | 65.6 | 66.4 |
| UCF101 | 84.0 | 85.5 | 87.2 | 87.6 |
| ESC50 | 78.9 | 84.1 | 84.6 | 84.9 |
| YouCookII | 17.9 | 20.7 | 24.2 | 23.1 |
| MSR-VTT | 14.1 | 14.6 | 15.1 | 15.2 |

Table 5: Top-1 accuracy of linear classification and R@10 of video retrieval vs. drop rate vs. inference GFLOPs in the VATT-MBS.

| Resolution/ | DropToken Drop Rate | | | |
| --- | --- | --- | --- | --- |
| FLOPs | 75% | 50% | 25% | 0% |
| $32 \times 224 \times 224$ | - | - | - | 79.9 |
| Inference (GFLOPs) | - | - | - | 548.1 |
| $64 \times 224 \times 224$ | - | - | - | 80.8 |
| Inference (GFLOPs) | - | - | - | 1222.1 |
| $32 \times 320 \times 320$ | 79.3 | 80.2 | 80.7 | 81.1 |
| Inference (GFLOPs) | 279.8 | 572.5 | 898.9 | 1252.3 |

Table 6: Top-1 accuracy of video action recognition on Kinetics400 using high-resolution inputs coupled with DropToken vs. low-resolution inputs.

# 5    Conclusion and Discussion

In this paper, we present a self-supervised multimodal representation learning framework based on Transformers. Our study suggests that Transformers are effective for learning semantic video/audio/text representations — even if one model is shared across modalities — and multimodal self-supervised pre-training is promising for reducing their dependency on large-scale labeled data. We show that DropToken can significantly reduce the pre-training complexity with video and audio modalities and have minor impact on the models' generalization. We report new records of results on video action recognition and audio event classification and competitive performance on image classification and video retrieval. Having these results, we still see some limitations in our work. Firstly, not all videos have organic audio or speech, while our approach depends on meaningful multimodal correspondences. Besides, the text modality currently consists of speech transcripts, which are noisy and sometimes sparse. Potential negative Societal Impacts are mainly concerned with applications. The models could be biased if one applies our approach to the multimodal videos that are not representative enough. Finally, our method is still demanding in computation, though we managed to avoid the need for human labels. Future work can improve upon these limitations.

## Acknowledgments and Disclosure of Funding

We would like to thank Min-Hsuan Tsai, Jean-Baptiste Alayrac, Andrew Audibert, Yeqing Li, Vidush Mukund, and the TensorFlow team for their help with codes, infrastructure, and insightful discussions.

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
