# A   Appendix

Appendix contains more detailed explanations about datasets (A.1) and the experimental setup (A.2) for both pre-training and downstream tasks. We also cover linear evaluation results compared to state-of-the-art (A.4) and an ablation study on the input parameters (A.5).

## A.1   Datasets

### A.1.1   Pre-training

Following [1, 59], we use HowTo100M [58] and AudioSet [33] to pre-train VATT. The former contains 1.2M unique videos, each providing multiple clips with audio and narration scripts resulting in 136M video-audio-text triplets in total. The narration scripts are extracted from speech audio using an off-the-shelf ASR. We use a subset of HowTo100M to comply with Youtube's policies, which results in having almost 1M unique videos and less than 100M clips. AudioSet consists of 10-second clips sampled from two million videos from YouTube. The dataset contains a variety of audio events with their corresponding video without any narration, so we do not have any text input from this dataset. We do not use any labels from the datasets. We uniformly sample clips from these datasets; a mini-batch in the pre-training contains samples from both datasets. In order to fill in the empty text in AudioSet, we feed a sequence of zeros to the text Transformer and exclude those samples from the MIL-NCE loss.

### A.1.2   Downstream

We evaluate the pre-trained VATT on a set of diverse, representative downstream tasks to test different aspects of the learned representations.

**Video action recognition:** We evaluate the visual representations on UCF101 [81] (101 classes, 13,320 videos), HMDB51 [52] (51 classes, 6,766 videos), Kinetics-400 [14] (400 classes, 234,584 videos), Kinetics-600 [15] (600 classes, 366,016 videos), and Moments in Time [61] (339 classes, 791,297 videos). Since UCF101 and HMDB51 are small datasets compared to the size of our model, we freeze the vision backbone and use its outputs to train a linear classifier. We use the split #1 results of the two datasets as a reference in our design exploration. For Kinetics-400, Kinetics-600, and Moments in Time, we fine-tune our vision backbone initialized from the pre-trained checkpoint.

**Audio event classification:** We use ESC50 [66] (50 classes, 2000 audio clips) and AudioSet [33] (527 classes, ∼2M audio clips) to evaluate our audio Transformer on audio event classification. We use ESC50 to train a linear classifier on top of the frozen audio Transformer. We use the split #1 results of this dataset as a reference in our design exploration. We also use AudioSet to fine-tune our audio backbone initialized from the pre-trained checkpoint.

**Zero-shot video retrieval:** We evaluate the quality of our video-text common space representations by *zero-shot* text-to-video retrieval on two of the most established datasets in this area: YouCook2 [109] and MSR-VTT [98] with 3.1k and 1k video-text pairs, respectively. We follow the same evaluation pipeline described in [1] and report the Recall at 10 (R@10).

**Image classification:** Although there exists a domain gap between images and the video datasets used for pre-training VATT, we test the learned vision Transformer in the image domain. We fine-tune the last checkpoint of the vision Transformer on ImageNet [22] with no modification to our architecture or the tokenization pipeline. We will elaborate on this in the sequel.

## A.2   Experimental Setup

### A.2.1   Inputs

During pre-training, we sample 32 frames at 10 fps for both pre-training datasets. For these frames, we randomly crop a temporally consistent spatial region whose relative area is in the range of [0.08, 1] and its aspect ratio in [0.5, 2]. These crops are then resized to $224 \times 224$, followed by a horizontal flip and color augmentation. The color augmentation follows [1] and randomizes brightness (max delta = 32/255), saturation (max delta = 0.4), contrast (max delta=0.4), and hue (max delta=0.2). We

clip values to ensure the RGB is in [0, 1]. The audio waveforms are sampled in sync with the video frames at 48kHz. Both video and audio inputs are normalized between [-1, 1] for numerical stability. We use patch sizes of $4 \times 16 \times 16$ and 128 for video and raw waveform tokenization, respectively. We use one-hot vectors to encode text sequences with the vocabulary size of $2^{16}$, which is the same as word2vec [60]. The resulting sequence retains a maximum of 16 words by either clipping or padding. We use DropToken with a drop rate of $50\%$ during pre-training. For video fine-tuning and evaluation, 32 frames with a temporal stride of 2 are sampled at 25 fps (2.56 seconds) with a crop size of $320 \times 320$ (with similar video augmentation during pre-training), and we do not drop any tokens. We do not change the input size for audio and text during evaluation.

### A.2.2 Network setup in VATT

We use the same Transformer architecture described in the main paper with various sizes shown in Table 7. We use the Medium model for our modality-agnostic variant (VATT-MA). For the experiments with modality-specific Transformers, we use the Small and Base models for the text and audio modalities, respectively, while varying the model sizes for the video modality. This results in 3 variants for the modality-specific video-audio-text backbones: Base-Base-Small (BBS), Medium-Base-Small (MBS), and Large-Base-Small (LBS).

| Model | Layers | Hidden Size | MLP Size | Heads | Params |
|-------|--------|-------------|----------|-------|--------|
| Small | 6 | 512 | 2048 | 8 | 20.9 M |
| Base | 12 | 768 | 3072 | 12 | 87.9 M |
| Medium | 12 | 1024 | 4096 | 16 | 155.0 M |
| Large | 24 | 1024 | 4096 | 16 | 306.1 M |

Table 7: Details of the Transformer architectures in VATT.

### A.2.3 Projection heads and contrastive losses

We use $d_{va} = 512$ and $d_{vt} = 256$ for the projection to the common spaces $\mathcal{S}_{va}$ and $\mathcal{S}_{vt}$, respectively. We normalize the vectors before calculating the NCE and MIL-NCE objectives and use the temperature of $\tau = 0.07$ and the weight of $\lambda = 1$ in the loss defined in the paper. We choose these values following the previously established practice [1]; we may achieve better results by varying these hyper-parameters.

### A.2.4 Pre-training setup

We pre-train VATT from scratch using Adam [46] with an initial learning rate of $1e$-4, 10k warmup steps, 500k steps in total, a batch size of 2048, and a quarter-period cosine schedule to anneal the learning rate from $1e$-4 to $5e$-5. In the exploration experiments, we use a batch size of 512 while keeping the rest of the training parameters the same. Our pipeline is implemented in Tensorflow (v2.4), and our models are trained for 3 days using 256 TPUs (v3).

### A.2.5 Video fine-tuning setup

For video action recognition, we use the SGD with a momentum of 0.9 and an initial learning rate of 0.005, 2.5k warmup steps, a batch size of 64, 100k steps in total, and a half-period cosine schedule to anneal the learning rate to 0. We use label smoothing with smoothing factor $\alpha = 0.1$. The video frame resolution is $320 \times 320$, which results in an increase in the number of positional encoding weights. This increase is due to the fact that, in the pre-train time, we have 8+14+14 positional encoding buckets, while 8+20+20 positional buckets are required to completely encode $320/16$ horizontal and $320/16$ vertical locations in fine-tune. To generate the new positional embeddings, we create a new set of positional encoding buckets by bi-cubic interpolation from the original buckets. After this step, we fine-tune the entire network, including the positional encoding buckets, end-to-end. We tried fixed positional embeddings (solely based on interpolation for the missing locations) and did not observe significant improvements. We uniformly sample 4 clips to cover the entire 10 seconds of the video and apply a standard 3-crop evaluation following [30]. We average the logits across the resulting 12 views before having the final class predictions.

## A.3 Audio fine-tuning setup

For audio event classification, we use the SGD with a momentum of 0.9, an initial learning rate of 0.2, 5k warmup steps, a batch size of 1024, 50k steps in total, and a half-period cosine schedule to anneal the learning rate to 0. We observe that increasing the effective receptive field improves the overall performance. We suggest that this might be due to the fact that the AudioSet annotations are multi-label and each event might occur in different temporal positions. Hence, we employ the duration of 6.4s with 24kHz sampling rate (153.6k total input samples). Similar to [49], we use mixup [101] on input-label ($x$-$y$) pairs in a mini-batch as below:

$$x = \alpha x_1 + (1 - \alpha)x_2, \qquad y = \alpha y_1 + (1 - \alpha)y_2,$$

where the input-label pairs are randomly sampled from a mini-batch, and the mixing rate $\alpha$ is sampled from a $\text{Beta}(5, 5)$ distribution. We also perform data balancing by penalizing the loss value of a sample with the inverse of the per-batch number of repetitive labels it carries. This is crucial for avoiding over-fitting since AudioSet has a long-tailed distribution, and a few dominant classes may disrupt the training [49].

### A.3.1 Image fine-tuning setup

We finetune the pre-trained VATT on ImageNet for 50 epochs with $384 \times 384$ input resolution, 512 batch size, SGD with momentum of 0.9, cosine learning rate decay with an initial learning rate of $8e$-2, and label smoothing of 0.1. No weight decay is used.

### A.3.2 Linear evaluation setup

We use a linear classifier with fixed backbones across all datasets and tasks. We observe that using matrix factorization on the classifier weight [75] leads to a more stable result across experiments. More specifically, we use a factorized weight $C = UV \in \mathbb{R}^{d \times c}$, where $U \in \mathbb{R}^{d \times n}$ and $V \in \mathbb{R}^{n \times c}$ are learnable weights. During training this classifier, we randomly choose a subset of the $n$ components in $U$ and $V$, hence leading to a low-rank classifier weight, $C$. The classifier weight, $C$, is trained using the Adam optimizer with a learning rate of $5e$-4, a batch size of 64, a total of 50k training steps, and a sampling rate of 10% on its $n = 128$ components.

### A.3.3 Zero-shot retrieval setup

For zero-shot text-to-video retrieval, we use the 1k split of MSR-VTT and the entire test split of YouCook2 as the pool for retrieval. We use $224 \times 224$ central crops for 32 frames with a temporal stride of 2 sampled at 25 fps. Since each input clip covers 2.56 seconds, and the full clip length is 10 seconds, we average the embeddings over 4 uniformly sampled clips before calculating the similarity with a text query's embedding. We $\ell_2$-normalize each vector to assure that a dot product results in the cosine similarity.

## A.4 Linear evaluation on frozen VATT

We also test VATT's ability to generalize to other datasets when the entire backbone is frozen. In this setting, we focus on the video and audio modalities and train a linear classifier on the outputs of the frozen backbones. In addition to the low-rank classifier (LRC) described in Section A.2, we also report the results of a SVM classifier following the same pipeline as [1]. Table 8 shows the performance of our model on three datasets. We observe that VATT does not outperform the best CNN counterparts in [1], and achieves comparable numbers to other baselines. This could suggest that VATT's backbones learn less-linearly-separable feature, especially given that the contrastive estimation head includes non-linear projections.

## A.5 Ablation study on input parameters

Since VATT takes raw multimodal signals as inputs, the choice of input size and how they are patched has a significant impact on the final performance. First, we alter the frame crop size and the number of sampled frames from each video clip while keeping the patch size fixed to $5 \times 16 \times 16$. Table 9 shows that using a small frame crop size and a larger number of frames hurts the video-related results, but it does not significantly change the audio classification numbers.

| METHOD | UCF101 | HMDB51 | ESC50 |
|---|---|---|---|
| MIL-NCE [59] | 83.4 | 54.8 | - |
| AVTS [50] | - | - | 82.3 |
| XDC [2] | - | - | 84.8 |
| ELo [67] | - | 64.5 | - |
| AVID [80] | - | - | **89.2** |
| GDT [65] | - | - | 88.5 |
| MMV [1] | **91.8** | **67.1** | 88.9 |
| VATT-Medium + SVM | 89.2 | 63.3 | 82.5 |
| VATT-Medium + LRC | 89.6 | 65.2 | 84.7 |
| VATT-MA-Medium + LRC | 84.4 | 63.1 | 81.2 |

Table 8: Linear evaluation results for video action recognition on UCF101 and HMDB51 and audio event classification on ESC50. MA refers to the Modality-Agnostic backbone.

| Frame Size | Patch Size | UCF | HMDB | YC2 | MSRVTT | ESC |
|---|---|---|---|---|---|---|
| $32 \times 224 \times 224$ | $4 \times 16 \times 16$ | **87.8** | **67.7** | **27.53** | **17.99** | **87** |
| $32 \times 200 \times 200$ | $5 \times 16 \times 16$ | 87.16 | 67.08 | 23.98 | 17.84 | 86.25 |
| $32 \times 224 \times 224$ | $5 \times 16 \times 16$ | 87.74 | 67.6 | 27.47 | 17.96 | **87** |
| $64 \times 224 \times 224$ | $5 \times 16 \times 16$ | 86.57 | 63.09 | 18.52 | 12.5 | 86.25 |
| $32 \times 224 \times 224$ | $8 \times 16 \times 16$ | 86.52 | 65.64 | 23.43 | 16.14 | 84 |
| $32 \times 224 \times 224$ | $8 \times 32 \times 32$ | 82.68 | 60.73 | 15.27 | 13.79 | 87 |

Table 9: Effect of video frame and patch size on downstream results.

Then, we keep the best frame size ($32 \times 224 \times 224$) and vary the video patch size. We find going beyond $4 \times 16 \times 16$ along either the time or spatial dimensions is not helpful. We avoid patches that are smaller than $4 \times 16 \times 16$ because of the significantly increaseed wall clock time in experiments.

Finally, we compare different audio patch sizes and perform an experiment using spectrograms, as opposed to the raw waveforms, as audio input. The goal is to see how the raw waveforms compare to the handcrafted spectrograms. We use the MEL spectrogram with 80 bins, the STFT length of 42 ms, and the STFT step of 21 ms following a similar setup in [1]. Tables 10 summarize the results, in which we observe that the patch size of 128 gives rise to the best waveform-based results, and using spectrogram does not lead to any conclusive improvement. The experiment with the spectrograms demonstrates that VATT is able to learn semantic representations from raw audios. To the best of our knowledge, this is the first time that raw audio waveforms are used for multimodal self-supervised learning.

| Input | Patch Size | UCF | HMDB | YC2 | MSRVTT | ESC |
|---|---|---|---|---|---|---|
| Waveform | 128 | **88.14** | **68.13** | 25.72 | **17.31** | **87.75** |
| Waveform | 256 | 87.74 | 66.1 | 24.19 | 16.55 | 83.75 |
| Waveform | 512 | 87.21 | 67.34 | **26.11** | 16.91 | 82.5 |
| Waveform | 1024 | 86.41 | 66.36 | 24.46 | 16.38 | 82.5 |
| Spectrogram | $16 \times 5$ | **88.3** | 67.52 | **26.62** | 16.86 | **88** |

Table 10: Effect of the audio input type and patch size on downstream results.