# OpenReview forum: "VATT: Transformers for Multimodal Self-Supervised Learning from Raw Video, Audio and Text"
_NeurIPS.cc/2021/Conference — NeurIPS 2021 Poster_

### Official Review · Reviewer_akAf · 2021-07-15

**Rating:** 6
**Confidence:** 4

**Summary:**

This paper describes a self-supervised pure transformer-based multimodal representation learning including video-audio-text, i.e., VATT. VATT consists of  linear projection of input for each modality, modality-specific or agnostic transformer encoder, and multimodal projection head. VATT uses NCE loss for video-audio feature learning and MIL-NCE loss for text-video features. Also, the authors proposed DropToken as an efficient method for training.
They employ HowTo1M and AudioSet datasets for pretraining and evaluate their VATT for diverse finetuning tasks including video action recognition, audio event classification, and zero-shot text-to-video retrieval. With intensive experiments and ablation studies, they provide promising results and constructive analysis.


**Ethical Concerns:**

No ethical concern

**Limitations And Societal Impact:**

No special societal impact.

**Main Review:**

### Strength
#### Originality
- It is novel to use audio for multimodal pretraining in addition to video and text.
- Pure transformer architecture-based video-audio-text learning is novel considering NeurIPS submission deadline.

#### Quality & Significance
- This paper is well-organized to easy to understand.
- Multimodal pretraining is a challenging but significant topic.

#### Clarity & Experiments
- Experiments are extensive and the results look promising on diverse downstream tasks.
- Analysis including feature visualization is helpful.

### Weakness
Although I like this paper, there are some rooms to be improved.
- Some core previous studies was missed in related work such as VideoBERT [Sun et al. 2019]. Also, The authors need to compare VideoBERT. Also, in related work, it is helpful to enhance the story by including recent studies on self-supervised learning from uncurated data in computer vision domains such as SEER [Goyal et al. 2021] and image-based self-supervised learning for videos [Feichtenhofer et al. 2021].
- Why did the author use word-level linear projection as text input instead of BPE tokenization, which is de facto in training LM? Please clarify this reason.
- The author employs multi-level common space projection with hierarchy. Is there any performance drop using pair-wise feature loss with plat-style? If then, how significant?
- For negative pairs, how about making matching inter video clips? for example, frames from video A and audio from video B.
- As the authors described in the paper, VATT model sizes are variable according to each modality encoder size and whether the encoder is modality-agnostic or not. At least, table 1, 2 and, 3 need to include model parameter sizes.
- In table 1, What data are used as the pretraining or scratch-training for other baseline models? It need to be clarified.
- How is the performance on Kinect-700? Including MMV, many recent work evaluated their model on Kinect-700.
- For DropToken, how is the effect of DropToken on each modality? Did the authors try to apply DropToken to a specific modality?
- Why VATT-MBS was compared instead of VATT-Base, unlike other tables? Considering the parameter size, I think that VATT-Base with the same batch-size and epochs is more suitable for comparison.
- How is the performance when increasing batch-size and epochs similar to MMV or MIL-NCE in table 4? As the authors argued the importance of batch-size and epoch, the results on large batch size and epochs with the same model will help to clarify the argument.
- Readability can be improved. For example, VATT-Base means VATT-BBS? In Table 4, "VATT-MBS" is used instead of "VATT-Medium". Consistent term usages are very helpful. Overall, the captions of Tables need to be self-contained with more explanation and term definition. Also, it is better to clarify the use of self-supervised ViT (79.9%) for better understanding.

#### References
- [Sun et al. 2019] A Joint Model for Video and Language Representation Learning. ICCV 2019.
- [Goyal et al. 2021] Self-supervised Pretraining of Visual Features in the Wild. arXiv 2021.
- [Feichtenhofer et al. 2021] A Large-Scale Study on Unsupervised Spatiotemporal Representation Learning. CVPR 2021.


**Time Spent Reviewing:**

10

---

> ### Author Response · Authors · 2021-08-10
> **Response to reviewer akAf**
>
> ## Q: Related Work
> We thank the reviewer for their suggestion. These are concurrent works and we will acknowledge their contributions in our related work section.
>
> ## Q: Comparing results to VideoBERT
> VideoBERT is a generative model based on pre-extracted features and vector quantization. However, VATT is an encoder which tries to minimize a cross-modal matching loss by semantic feature extraction in a common space. To the best of our knowledge, VideoBERT reports numbers on only two tasks: 1. Action Classification through generation, 2. Video Captioning. Hence, we did not find a shared downstream task to compare our model to VideoBERT. Future work can study the autoregressive generation on top of VATT’s encoder.
>
> ## Q: Why word-level linear projection?
> We would like to clarify that we do not have a word-agnostic linear projection. We do follow the established practice in the LM training by having a vocabulary-size embedding matrix that projects a large one-hot encoding to an embedding space. We will make sure this is clear in the paper. For tokenization, we tried BPE and WordTokenization and achieved better results using WordTokenization. This might be because of the limited language structure in the HowTo100M dataset. Since the text in this dataset is based on ASR, we argue that the model learns to attend to the related object name mentions in a sentence. By visual inspection, we also saw that there is no significant language structure in the sentences generated by the ASR system in this dataset and they are very noisy. This is one of the reasons that MMV uses a very simple average pooling over the word representations and still achieves good results.
>
> ## Q: Is there any performance drop using pair-wise feature loss with plat-style?
> Yes. We observed that there is a performance drop, especially on the zero-shot retrieval results. We explored different options and reported the best in the paper. We will add these numbers to the appendix of the paper.
>
> |  Model      |  UCF-101  | HMDB-51 | YouCook2 |
> | ------------- |:-------------:|:-------------:|:-------------:|
> |  Plat-style |    84.6      |     63.6     |    32.4      |
> | Hierarchical |  **88.1**  |  **66.9**  |  **38.7**  |
>
> Note: Since these are from parameter exploration, batch size is 512 hence the numbers are different from the ones in the paper.
>
> ## Q: How about making matching inter video clips?
> The current formulation samples negatives from any clips. A modality from video clip A is compared to another modality from video clip B and the assumption is that they are negative to each other. Since we do not hold or use any labels for the pre-training datasets, we cannot sample positive pairs from inter clips. We hope this explanation addresses the reviewer’s question.
>
> ## Q: Parameter sizes in Tables 1-3
> We appreciate the reviewer’s suggestion. We will add the parameter count for all models in our tables. We have included the computational cost for the models in Table 1 and will include that one as well for the rest of the tables.
>
> ## Q: Pre-training data in Table 1
> Unless otherwise noted, we use the latest checkpoint of the pre-trained VATT in all of our downstream finetuning. We briefly mention in lines 243-244 that training from scratch gives very bad results for downstream tasks. This is the major issue with Transformer-based models. VATT provides a straightforward pre-training pipeline for a standard Transformer to resolve this issue. As we have shown in Tables 1-2, VATT even sets new records on 6 datasets.
>
> ## Q: Results on Kinetics-700
> We thank the reviewer for their suggestion. We finetuned our model on Kinetics-700 and observed new records on this dataset as well.
>
> |      | Kinetics-700 Top-1 Acc |
> | ------------- |:-------------:|
> | SlowFast-R152      | 71.6     |
> | MoviNet-A6 (SoTA)| 72.4     |
> | VATT-Large      | **72.7**    |
>
> ## Q: DropToken on modalities
> Since there is a lot of redundancy in raw video and audio inputs, we only apply DropToken on these modalities. DropToken is aimed to reduce computational cost by pruning redundancy. Hence, we did not apply it on text because text is already short (16 tokens) and does not contribute significantly to the computational cost. However, we will explore it on Text in our future work.
>
> ## Q: Performance of retrieval when increasing batch size and epochs
> We have mentioned this in the paper. We trained our model with a batch size of 8192 and only on 6 epochs. We observed a significant improvement for YouCook2 but a slight degradation for MSR-VTT.
>
> |                 | Batch Size | Epochs | YouCook2 | MSR-VTT |
> | ------------- |:-------------:|:-------------:|:-------------:|:-------------:|
> | MIL-NCE  |     8192    |        27      |   **51.2**   |  **32.4**  |
> | MMV        |     4096    |         8       |     45.4     |     31.3     |
> | MMV (re-trained)|     2048    |         4       |     29.3     |     21.1     |
> | VATT        |     2048    |         4       |     45.5     |     29.7     |
> | VATT        |     8192    |         6       |   **51.2**   |     29.2     |
>
> ## Q: VATT’s naming
> We apologize if this has caused confusion. All of the downstream models are initialized from the last checkpoint of the full multimodal VATT pre-training. Furthermore, since each modality in the modality-specific setting has its own size, we refer to that modality’s size when finetuning on downstream tasks. For example, we only used the Base size for Audio in all experiments, hence referring to it as VATT-Base in Table 2. For Video, we found it useful to increase the size and trained three variants: Large-Base-Small, Medium-Base-Small, Base-Base-Small (referring to Video-Audio-Text backbones respectively). That’s the reason that we see three sizes in Table 1. In Table 3, to be fair we only reported the Base variant’s results. Finally, in Table 4 since it is a multimodal evaluation we refer to the full size MBS.
> All modality-agnostic settings only have a Medium size, hence referring to them as MA-Medium (Modality-Agnostic-Medium).

---

> > ### Comment · Reviewer_akAf · 2021-09-01
> > **Thank you for the efforts**
> >
> > I carefully read the other's reviews and the author feedback. First, I thank the authors for their efforts. The authors alleviated my main concerns. However, I agree with other reviewers' concerns such as technical contributions. Even if percevier [Jaegle] is not the previous work that should be explicitly considered due to its published venue (ICML, arXiv in March), the main contribution of unified transformer architecture might be weaken. For videoBERT, although VATT is remarkably different from videoBERT, the authors need to clarify this difference in related work. Overall, I adjucted my score to 6. If this paper is accepted, please clarify the terms and model-names for better readability.
> >
> > [Jaegle 2021] Perceiver: General Perception with Iterative Attention. ICML 2021.

---

> > > ### Author Response · Authors · 2021-09-02
> > > **Response to reviewer akAf's follow-up**
> > >
> > > We really appreciate the reviewer for carefully following up and increasing the rating to 6.
> > >
> > > We would like to emphasize that the major goal of Perceiver is a unified pipeline for **raw inputs** processing with linear attention.
> > > This is different from the major contribution of VATT which focuses on sharing a single backbone across different modalities. The Perceiver does not consider training one backbone (set of weights) on multiple modalities at once. Instead, they train one architecture multiple times for multiple datasets and tasks.
> > >
> > > We will make sure that we include the difference compared to videoBERT and we will also clarify our model naming to avoid the confusion.
> > > We appreciate the reviewer's concerns.

---

### Official Review · Reviewer_6L5v · 2021-07-16

**Rating:** 4
**Confidence:** 4

**Summary:**

This paper provides a framework for self-supervised multimodal representations from unlabeled data with Transformer backbones. The authors study two styles of Transformer encoders: modality-specific or modality-agnostic.  Finally, they achieve good performance on various downstream tasks including action recognition, audio event classification, text-to-video retrieval and image classification.

**Main Review:**

- This paper is well written and the topic of this paper is valuable.

- The novelty of this paper might be limited. The framework of self-supervised multimodal representation learning is almost as same as MMV [1] except replacing CNN backbones with Transformers.  DropToken technique does not show many advantages especially over training with lower resolution (79.9 acc. with 548.1 GFLOPS v.s. 80.2 acc. with 572.5 GFLOPS in Table 6).

- It is interesting to study the modality-agnostic encoder and the results are encouraging. However, it is not the main part of this paper. It would be nice to take a more detailed analysis and prove that a modality-agnostic encoder can achieve competitive or better performance than a modality-specific one.

- The performance comparisons with previous multimodal representation learning methods (MIL-NCE, MMV) are missing. Authors should following previous papers that evaluating on UCF and HMDB and report the results.

- The detail of DropToken is confusing. Is the DropToken used in both the training and testing phases? Is it used when transferring to downstream tasks or only during pre-training?


**Time Spent Reviewing:**

10

---

> ### Author Response · Authors · 2021-08-10
> **Response to reviewer 6L5v**
>
> ## Q: Novelty and Contribution
> This work has major distinctions from similar works in the literature.
> 1. Compared to MMV, our model operates on full-resolution raw inputs and requires no pre-processing (other than augmentation for training). In our experiments with MMV, we always faced problems for data processing bottlenecks (data pipeline falling behind the training loop). Furthermore, VATT achieves superior performance compared to MMV on a variety of downstream tasks.
> 1. Compared to ViT’s line of research, VATT is the first end-to-end multimodal Transformer in the literature. VATT sets new records on a variety of datasets, and does not focus on one modality or one task only and reformulates Transformers for Video, Image, Audio, and Text all at once.
> 1. VATT explores the idea of sharing one architecture, hence one weight, across different modalities for the first time. We show promising results and define a new line of research around multimodal understanding. We argue that we should distance ourselves from ad-hoc modality-specific network design and focus on training an integrated end-to-end pipeline that operates on raw inputs. Transformers enable us to do so.
> 1. We were able to train the same model on a different domain (image) with no network modification. VATT is the first major proof of concept toward a direction in which a task-agnostic set of weights could be used for a wide range of downstream tasks and modalities.
>
> ## Q: DropToken
> We introduce DropToken to have direct control over the number of FLOPs during training. One of the immediate benefits of this approach is that we can easily pre-train a very large model with limited resources and finetune it using the same high resolution on a smaller downstream dataset. In our experiments, we were not able to train the largest setting without DropToken. Yet, the largest pre-trained model gives the best results in the downstream tasks. This wouldn’t have been possible without DropToken. Table 6 shows the very minimal effect that DropToken has on the accuracy while significantly reducing the FLOPs. For example, it shows that with almost halving the FLOPs, we only lose 0.6% accuracy. To the best of our knowledge, this method is the first of its kind in the literature.
>
> |       | GFLOPs | Top-1 Accuracy |
> | ------------- |:-------------:|:-------------:|
> | 32x224x224     |548.1|  79.9  |
> | 32x320x320 + DropToken|279.8|  79.3  |
>
> ## Q: Further analyses on modality-agnostic setting
> We appreciate the reviewer’s suggestion. We included the modality-agnostic’s results in almost all of our evaluations to show how it compares to the modality-specific setting. Thanks to the reviewer’s suggestion, we performed another analyses to understand how the modality-agnostic backbone activates w.r.t different modalities. The figure below shows that the model is selective in the earlier layers. For example, text modality activates the model very early (most probably because of the difference in semantic level of representation), while the more redundant modalities (e.g. audio and video) activate the model later in the middle layers. We can see that interestingly the model activates to all modalities (describing the same concept) in the later layers that the level of semantic representation is much higher. We will include this interesting analysis in the paper.
>
> Figure: [Modality-Agnostic GeLU activations](https://i.ibb.co/DpRVkZt/Screen-Shot-2021-08-10-at-4-30-48-PM.png)
>
> ![](https://i.ibb.co/DpRVkZt/Screen-Shot-2021-08-10-at-4-30-48-PM.png)
>
> ## Q: Results on UCF and HMDB
> We retrained MMV using the same framework and achieved the following results. We can see that VATT significantly outperforms MMV in all metrics. VATT not only outperforms MMV on linear probing and zero-shot retrieval, it also achieves new records on finetuning with action recognition and audio classification datasets. Please note that the numbers are different from the ones reported in the MMV paper because of HowTo100M shrinkage. The current HowTo100M online has ~20% less number of samples compared to the ones from last year. The exact numbers could not be reproduced even with hyperparameter search.
>
> |  Model      | Kinetics-400  | Kinetics-600  | UCF-101  | HMDB-51 |   ESC-50 | YouCook2 | MSR-VTT |
> | ------------- |:-------------:|:-------------:|:-------------:|:-------------:|:-------------:|:-------------:|:-------------:|
> | MMV (TSM)        |    76.7      |    78.3      |    84.6      |    63.6      |    81.2      |    29.3      |    21.1      |
> | VATT (Transformer)        |  **82.1**  |  **83.6**  |  **89.6**  |   **65.2**  |  **84.7**  |  **45.5**  |  **29.7**  |

---

> > ### Comment · Reviewer_6L5v · 2021-09-02
> > **Thanks for the rebuttal**
> >
> > After carefully reading the author rebuttal and other reviews, I appreciate the authors' effort on preparing the rebuttal. I agree with other reviewers that this paper lacks original novelty and might need much more computational cost for training.  The result is interesting for the research community. However, the method might be not practical for groups with limited computational resources, such as university lab. I think this paper is hard for most of research group to reproduce and conduct follow-up research. Therefore, I keep my original rating.

---

> > > ### Author Response · Authors · 2021-09-02
> > > **Response to reviewer 6L5v's follow-up**
> > >
> > > We thank the reviewer for carefully following up on the rebuttal discussion.
> > >
> > > We would like to emphasize that the computational cost for training VATT is less than MMV and it is comparable to many of the recent influential Transformer-based work in the field (e.g. ViT, ViViT, etc.). For example, a forward call for video using ViViT requires 876 GFLOPs which is much higher than VATT's 548 GFLOPs. Similarly, Perceiver requires 400 GFLOPs for forward call on a 224x224 still image, while a similar forward call with VATT requires only 68 GFLOPs.
> > >
> > > As we mentioned in our response to reviewer PWef's follow-up, the MMV paper only reports their computational cost when significantly downsampling the input data, which results in a 96 TPU days cost. Performing the same downsampling in VATT results in only 50 TPU days which is almost half the number for MMV. Transformers are indeed more efficient than large ConvNets e.g. TSMx2 or R50x2.
> > > We will add this information in the paper to clarify that a low-resource setting still results in comparable numbers.
> > >
> > > Moreover, we have introduced **DropToken** which significantly reduces the computational cost and enables training this pipeline on almost any computational setting.
> > >
> > > We have included all of these options in our codebase and made sure that the code is fully re-usable and configurable for the research community considering their computational resources.

---

### Official Review · Reviewer_PWef · 2021-07-16

**Rating:** 4
**Confidence:** 4

**Summary:**

This paper concerns multimodal self-supervised learning. It adopts the now ubiquitous contrastive learning framework to learn video, audio, text representation from each raw signal in joint embedding spaces. The proposed architecture heavily relies on MMV [1], with a twist on Visual/Audio Transformer encoders and modality-agnostic Transformers. Comprehensive experimental results are conducted on four diverse video(-text) tasks across ten datasets.

**Limitations And Societal Impact:**

Appear to be.

**Main Review:**

This paper is sending some mixed messages. On the one hand, it demonstrates Transformers' decent capability on learning from raw visual/audio signals, in the context of multimodal self-supervised learning. Also, the possibility of unifying encoder across modalities through a modality-agnostic framework. On the other hand, this unique capability is at the cost of larger models, more training resources, and sometimes performance degradation.

Since the proposed methods **heavily** rely on the previous work MMV [1], missing this important baseline in many result tables is a major flaw (despite that MMV is not originally evaluated on these highly relevant datasets) and hinders a fair comparison between CNN-based methods and Transformer-based methods. On the few datasets that have MMV results reported (e.g., Tab. 4, Tab. 2 in Supp.), the proposed methods overall underperform the baseline, sometimes significantly, despite using much larger models (2-4x) and computation resources (over 2x TPU days). This puts a doubt on the effectiveness of the proposed method.

Besides, the reported model variants are inconsistent throughout the paper (e.g., in Tab. 2 and 3, VATT-Base while in Tab. 4 VATT-Medium). In Tab. 1, columns indicating #params, #training data, modality should be added for better comparison.

**Time Spent Reviewing:**

3

---

> ### Author Response · Authors · 2021-08-10
> **Response to reviewer PWef**
>
> ## Q: Cost of the model
> Surprisingly, the cost of training this model is less than MMV’s best performing setting (TSM-50x2). The original MMV paper only reports the average cost across different large-scale settings (e.g. SGD, etc.). The table below shows a comparison of training cost for each of the largest settings for VATT vs. MMV:
>
> |  Model      | Params (M) | FLOPs |   TPU Days |
> | ------------- |:-------------:|:-------------:|:-------------:|
> | MMV        |    **142**      | 1030 GFLOPs |    1280      |
> | VATT        |    280      | **785** GFLOPs |  **768**  |
>
> As argued in the community, the number of parameters is not directly coupled with cost of training, especially when it comes to Transformers. Memory consumption is another important factor. Unlike VATT, MMV cannot take a full-resolution input because of OOM error. Following the official MMV’s codebase, we had to downsample on time and space and accumulate on the channels dimension to be able to train MMV. This trick (a.k.a space-time to depth) has been enforced in the official MMV code.
>
>
> ## Q: VATT’s finetuning results compared to MMV
> We appreciate the reviewer’s suggestion. We fine-tuned MMV’s vision backbone on Kinetics400, Kinetics600, Kinetics700, and MiT and achieved the following results:
>
> |  Model      | Kinetics-400  | Kinetics-600 |   Kinetics-700 | MiT |
> | ------------- |:-------------:|:-------------:|:-------------:|:-------------:|
> | MMV        |    76.7      |    78.3      |    68.2      |    31.4      |
> | VATT        |  **82.1**  |   **83.6**  |  **72.7**  |  **41.1**  |
>
> It can be clearly seen that the ConvNet baseline fails to achieve comparable results. We pre-trained MMV on the full dataset with the exact parameters reported in the original paper and fine-tuned it on video action recognition with 5x more epochs than ours and the best performing hyperparameters (found using grid search). It is worth noting that, for MMV, a full 32-frame video with a resolution of 320x320 was not even fitting on the TPUs easily, and we had to scale to more TPUs.
>
> ## Q: VATT’s linear probing and retrieval results compared to MMV
> We indeed adapted the MMV’s official (company-wide) code to train their model. Re-training their model does not give the same results, mainly because of HowTo100M’s constant shrinking problem. We have to follow a wiping policy and are only allowed to use the current public videos for training. Hence, we had almost 20% less data compared to the original MMV published in NeurIPS 2020. We will include the reproduced numbers in the paper to avoid any confusion. Please find the reproduced numbers below, which shows the superior performance of VATT compared to MMV when trained on same amount of data on the same framework:
>
> |  Model      | UCF-101  | HMDB-51 |   ESC-50 | YouCook2 | MSR-VTT |
> | ------------- |:-------------:|:-------------:|:-------------:|:-------------:|:-------------:|
> | MMV        |    84.6      |    63.6      |    81.2      |    29.3      |    21.1      |
> | VATT        |  **89.6**  |   **65.2**  |  **84.7**  |  **45.5**  |  **29.7**  |
>
> ## Q: VATT’s naming
> We apologize if this has caused confusion. All of the downstream models are initialized from the last checkpoint of the full multimodal VATT pre-training. Furthermore, since each modality in the modality-specific setting has its own size, we refer to that modality’s size when finetuning on downstream tasks. For example, we only used the Base size for Audio in all experiments, hence referring to it as VATT-Base in Table 2. For Video, we found it useful to increase the size and trained three variants: Large-Base-Small, Medium-Base-Small, Base-Base-Small (referring to Video-Audio-Text backbones respectively). That’s the reason that we see three sizes in Table 1. In Table 3, to be fair we only reported the Base variant’s results. Finally, in Table 4 since it is a multimodal evaluation we refer to the full size MBS.
> All modality-agnostic settings only have a Medium size, hence referring to them as MA-Medium (Modality-Agnostic-Medium).

---

> > ### Comment · Reviewer_PWef · 2021-09-01
> > **Follow-up**
> >
> > Note that the initial review is by no means encouraging adding more experiments in the rebuttal, but rather stating facts. This is unfair to other submissions; nor fair to the baseline method MMV (rushing out pre-training at such a short period of time is surely not ideal; with 20% (?) less data, the model would need a fresh hyper-param tuning, such as stronger regularization than the original setting). Having this fair data setting up-front, along with proper tuning to the baseline method would be instead appreciated. Setting aside the reproduced, compromised results on MMV, the reviewer is still not convinced the paper is sending a clear message that could facilitate further study or exemplifying fair comparisons between proposed work vs. existing work for future research.
> >
> > Please also add a reference for the training time on MMV. From the original paper "... training TSM-50 takes 3 days on 32 Cloud TPUs." which does not match 1280 TPU days reported here.

---

> > > ### Author Response · Authors · 2021-09-02
> > > **Response to reviewer PWef**
> > >
> > > We understand the reviewer’s concern and appreciate that they mentioned the fairness and other details related to MMV.
> > >
> > > We started working on this paper by first re-generating MMV’s results. Hence, we did perform all hyper-parameter tuning related to the shrunk data. Also, we utilized the official internal codebase that the original MMV authors had implemented. We only changed the framework version (from TF1 to TF2). As we mentioned in the rebuttal, training an MMV with a resolution of 32x224x224x3 (exact same resolution as VATT’s) takes significantly longer time, which is why we reported 1280 TPU days for MMV in the rebuttal. The MMV’s actual code takes a 32x200x200x3 input, downsamples the spatio-temporal information multiple times and stacks them to the channel dimension (resulting in 16x100x100x24) and feeds to the stem layer of the convolutional layer. This is the reason that they report 32 TPUs for 3 days, which is 96 TPU days. However, this trick damages the downstream results. Doing the same downsampling in VATT does the same reduction in the number of TPUs, which again results in much less number of TPU days compared to MMV. Taking a 32x200x200x3 input, downsampling it to 16x100x100x24 similar to MMV and feeding it to VATT results in a total of ~50 TPU days.
> > >
> > > Hence, in any equal situation the computational cost of VATT is less than MMV, thanks to the efficient XLA kernels used in a typical Transformer and also due to the fact that we share one architecture across different modalities.
> > >
> > > Even though we do get better results compared to MMV, we would like to emphasize that the focus of this paper is not by any means improving MMV. In this paper we try to unify and simplify the multimodal understanding pipeline by utilizing an input-agnostic architecture (Transformer). This paper is the first that shows there exists a single model (or one set of weights) that can process and perceive different modalities without separate modality-specific training.
> > >
> > > On a side note, we would also like to emphasize that the computational cost of VATT is less than many of the recent influential Transformer-based work in the field (e.g. ViT, ViViT, Perceiver, etc.). For example, a forward call for video using ViViT requires 876 GFLOPs which is much higher than VATT's 548 GFLOPs. Similarly, Perceiver requires 400 GFLOPs for forward call on a 224x224 still image, while a similar forward call with VATT requires only 68 GFLOPs.

---

> > > > ### Comment · Reviewer_PWef · 2021-09-03
> > > > **Clarification**
> > > >
> > > > Thanks for the reply. When you say "... share one architecture across different modalities..." in paragraph 3, are you referring to the modality-agnostic model? If so, what aspect of computational cost do you mean? The modality-agnostic model could indeed save parameters, but should not have any major impact on FLOPs and training time (as the forward pass remains the same for each modality), if the reviewer understands correctly. Please correct me if I am wrong.
> > > >
> > > > Also, the modality-agnostic idea is based on a strong assumption that encoders for different modalities should share the same complexity, which is not necessarily true as the vision encoder typically benefits more from a larger model than audio/text (as in this work and also MIL-NCE and MMV).
> > > >
> > > > I maintain my original rating Reject as the final rating, as the rebuttal/clarification does not change the original judgment of this paper in terms of its technical contribution and experiment fairness.

---

> > > > > ### Author Response · Authors · 2021-09-03
> > > > > **Response to PWef's Clarification Request**
> > > > >
> > > > > ### One architecture and one backbone
> > > > > By "...share one architecture..." we mean Transformer, since Transformers are more efficient on the advanced accelerators because of their MatMul-heavy operations. In our work, both modality-specific and modality-agnostic settings use one architecture, Transformer.
> > > > >
> > > > > On the other hand, the modality-agnostic has a better training time because we also **share backbone**. Although the forward call is still the same, the number of gradients is 1/3 of the modality-specific setting. Which means less updates per step and less memory consumption for holding the gradients.
> > > > >
> > > > > ### Assumption for encoders in the modality-agnostic setting
> > > > > The assumption of **same complexity** is not necessarily true and we do not put such assumption. Indeed, in our analyses we learned that the model forwards certain activations for specific modalities through the skip connection. Please take a look at this figure: [Modality-Agnostic GeLU activations](https://i.ibb.co/DpRVkZt/Screen-Shot-2021-08-10-at-4-30-48-PM.png)
> > > > >
> > > > > We observe that the model is activated in certain layers for certain modalities. This is an interesting observation and we believe that this could push the research community toward more unification and integration, which encourages against ad-hoc designs and settings.
> > > > >
> > > > > That is the reason that we believe the modality-agnostic idea is worth an attention in the field. Especially, we believe that this work is a great proof of concept for the new line of research, Foundation Models [1]. An ideal example of a foundation model is a unified and integrated model which can understand multiple modalities for multiple tasks (please refer to Fig. 2 in [1]). This is what our modality-agnostic VATT does and it is for the first time in the field that we observe such effort.
> > > > >
> > > > > [1] Bommasani, Rishi, et al. "On the Opportunities and Risks of Foundation Models." arXiv preprint arXiv:2108.07258 (2021).

---

### Official Review · Reviewer_taBL · 2021-07-17

**Rating:** 6
**Confidence:** 3

**Summary:**

This submission is about multi-modal representation learning from unlabeled data (self-sup.) using Transformer. This work deals with video, audio, and text data as multi-modal data. The approach to tackle representation learning is based on Transformer + contrastive learning.
The contrastive learning in this work is interesting in that 1) noisy association is considered in multiple instances learning way [64], and 2) the different level of semantic granularity is carefully considered.
Also, the authors adopt a simple DropToken to improve efficiency.

The training requires very large-scale training using HowTo100M and AudioSet. The authors demonstrated its effectiveness on action recognition, audio event classification, zero-shot video retrieval, image classification.

Multi-modal data is encoded by independent feature extractors, which is a small difference from other Vision Transformers. Naturally, the authors extend the positional encoding to 3D positional encoding (horizontal, vertical, and temporal axes) with relativeness.

Overall, the technical contribution is a bit weak, but this submission contains interesting analyses that are worthwhile to report in the community and also achieve state-of-the-art in downstream tasks.

**Limitations And Societal Impact:**

The authors discuseed the requirement of large computation, and model bias due to the use of uncurated multi-modal videos, which are reasonble concerns.


**Main Review:**


Pros
- Trendy topic with good results
- Pointing out different levels of semantic granularity
- Proposing DropToken to improve training efficiency
- Interesting experiment about the ImageNet finetuning results with the model pre-training video
- Extensive evaluation

Cons
- Weak technical contribution, and company-level large-scale training requirement
- Composition of existing developments

<Comments>

This work could be categorized as an analysis-oriented paper.

* Technical contribution: This work can be viewed as a similar line of research to the contrastive learning-based Vision Transfomer representation learning, e.g., [18]. In this regard, this work also can be viewed as a straightforward extension to multi-modal-based self-supervised data. The consideration of semantic granularity difference and Droptoken is only different from the previous work, but this reviewer feels that technical contribution is not significant.

* Semantic granularity: This reviewer concurs with the authors' argument on the different levels of semantic granularity in multi-modal learning. This reviewer thinks this must have been considered in all multi-modal learning research. At least, good to see the emphasis on it in this work.

* Showing a possibility of the existence of a unified model in a multi-modal regime. Nice.


** Question: Considering the out-of-context works on the recent MLP based backbones, what would be the true benefit of Transformer in this work? This reviewer does not see whether Transformer is a truly important component in the multi-modal learning regime like this work, other than large capacity.
Also, the submission does not include much about the necessity or motivation of the Transformer in multi-modal learning, but just mentioning no one did yet.\
What is the motivation of Transformer in the authors' opinion?

It would be more thorough to consider the motivation of the Transformer in this context.


======================== After rebuttal ========================

The authors have addressed most of the comments. However, the concerns on the technical contribution and the position of this submission are not adequately addressed, which are shared across the majority of the reviewers.

For this reason, I downgrade the score.
This reviewer still thinks that the benefits of accepting this work would be slightly higher than that of rejecting it to the community.



**Time Spent Reviewing:**

5

---

> ### Author Response · Authors · 2021-08-10
> **Response to reviewer taBL**
>
> ## Q: Technical Contribution
> The base spirit of this paper is similar to the standard ViT where the possibility of replacing the ConvNet backbone with a Transformer is assessed. However, we take major distinctive steps further for the first time in the literature:
> 1. Operating on raw inputs for three modalities w/o ad-hoc pre-processing (e.g. avoiding pre-extracted features, spectrogram, etc.). The original MMV work lacks this important transition. Also the original ViT (and many related work) only focus on one modality and one task.
> 1. A modality-agnostic pipeline where one set of weights is utilized for perceiving three modalities.
> 1. Wide multimodal evaluation and setting new records on 6 major video action recognition and audio event classification datasets.
> 1. Approaching the issue of domain gap between video and image tasks (i.e. using the same model trained on video data but for image task). This is a promising direction for the future work in the field toward unification of architectures and pre-training.
> 1. Introducing DropToken, an effective method for reducing computational complexity during training of a large Transformer model. Thanks to DropToken, we were able to pre-train very large models (~400M) and finetune them on downstream tasks and achieve new records. A similar ConvNet-based setting cannot be trained on the same input resolution.
>
> ## Q: The benefit of the Transformer architecture
> One of the immediate benefits of Transformer is the possibility of having one architecture and sharing that architecture across multiple modalities. Before this work, all work have been trying to rely on ad-hoc designs for modality-specific tasks, datasets, etc. We introduce the modality-agnostic approach which (thanks to Transformer) shares one architecture and even one weight across different modalities. This not only improves the overall design of the multimodal-multitask pipeline but also significantly improves the results in downstream tasks.
> Below please find a comparison between the VATT and MMV (our implementation) on 1. Finetuning, 2. Linear probing, and 3. Zero-shot retrieval
>
> |  Model      | Kinetics-400  | Kinetics-600  | UCF-101  | HMDB-51 |   ESC-50 | YouCook2 | MSR-VTT |
> | ------------- |:-------------:|:-------------:|:-------------:|:-------------:|:-------------:|:-------------:|:-------------:|
> | MMV (TSM)        |    76.7      |    78.3      |    84.6      |    63.6      |    81.2      |    29.3      |    21.1      |
> | VATT (Transformer)        |  **82.1**  |  **83.6**  |  **89.6**  |   **65.2**  |  **84.7**  |  **45.5**  |  **29.7**  |
>
> ## References
> *ViT:* Dosovitskiy, Alexey, Lucas Beyer, Alexander Kolesnikov, Dirk Weissenborn, Xiaohua Zhai, Thomas Unterthiner, Mostafa Dehghani et al. "An image is worth 16x16 words: Transformers for image recognition at scale." In ICLR 2020.
>
> *MMV:* Jean-Baptiste Alayrac, Adrià Recasens, Rosalia Schneider, Relja Arandjelovic, Jason Ramapuram, Jeffrey De Fauw, Lucas Smaira, Sander Dieleman, and Andrew Zisserman. Self339 supervised multimodal versatile networks. In NeurIPS, 2020.

---

> > ### Comment · Reviewer_taBL · 2021-08-25
> > **Reply to the authors' response**
> >
> > Thanks for the authors' response.
> >
> > However, unfortunately, the authors' responses are off the points against this reviewer's initial concerns.
> >
> > - Technical contribution\
> > : The points the authors listed are already well-received (especially, the finding of the possibility of the unification is well-taken). But this reviewer pointed out the "technical" contribution. Answers 1,2,3, and 4 are not the technical contributions.  This concern is critical because this concern is shared across a majority of the reviewers.
> >
> > - Benefit of Transformer architecture\
> > : This reviewer asked about the importance of the Transformer architecture over the recent alternatives of MLP-based architectures. The answers were all off. At a higher level, the motivation why Transformer is important in this work is required to be discussed.
> >
> > I'd like to ask the authors to deal with the common concerns across the reviewers properly.
> > At this point, these unresolved concerns may lower the score of this reviewer.
> > However, despite these concerns, this reviewer still thinks that the benefits of accepting this work would be a slightly higher than that of rejecting it to the community. More discussion across the reviewers is needed.

---

> > > ### Author Response · Authors · 2021-08-27
> > > **Official response to Reviewer taBL**
> > >
> > > We thank the reviewer for the constructive feedback.
> > >
> > > ### Technical contribution:
> > > To the best of our knowledge, ​​we are the first to show there exists a strong, unified model for multimodal (video, audio, text) perception. This is the major contribution of our paper and can be seen as a major step toward a unified foundation model [1].
> > >
> > > Proposing a new architecture for multimodal learning is interesting but not the focus of this work. Instead, we focus on minimizing the model architectural changes in order to unify with established achievements in Transformers for NLP and Vision. Therefore, we use the same exact model architecture as BERT and ViT so that the learned model can be easily applied to various downstream tasks and codebases for different modalities.
> > >
> > > Moreover, there are non-trivial design choices and empirical studies required to make the whole pipeline work. For example, DropToken is a novel component that is essential in our framework to reduce the computational complexity. Without DropToken, we were not able to train large multimodal models on multimodal data with high redundancy.
> > >
> > > Therefore, we believe that our work made significant contributions to the multimodal research community. We hope that the simplicity and flexibility of our pipeline and the convincing empirical evidence would foster emerging new research in this field, which is first of its kind.
> > >
> > > ### Benefit of Transformer architecture (MLP vs. Transformer):
> > > Upon observing the unprecedented success of Transformer architectures in both NLP and Vision, especially the architectures' capability of scaling up, we believe this is the right time to explore their usefulness to multimodal learning. In contrast, the MLP-based architectures (which are considered concurrent work to ours) are promising yet still lack signals to justify their use across different modalities. For example, they generally underperform Transformers on ImageNet, and it is unclear where they stand in NLP tasks. We agree with the reviewer that it is worth further exploring the MLP-based architectures, and yet it might be better to start with single-modality models first (e.g., how they will transform the NLP tasks).
> > >
> > > [1] Bommasani, Rishi, et al. "On the Opportunities and Risks of Foundation Models." arXiv preprint arXiv:2108.07258 (2021).

---

> > > > ### Comment · Reviewer_GNEo · 2021-08-31
> > > > **Agree with reviewer taBL, with additional comments**
> > > >
> > > > Thank authors for replying. I have some comments following this discussion.
> > > >
> > > > 1. I agree with reviewer taBL that the motivation of the transformer architecture should be discussed.
> > > > I don't agree with the authors the "benefits of the transformer" mentioned above (e.g. success, scaling-up, etc.) is the core benefits of the transformer here.
> > > > I think the key reason that building a unifying model with transformer architecture is beneficial is because the transformer is input-dependent.
> > > >
> > > > 2. I don't agree with the claim ***"​​we are the first to show there exists a strong, unified model for multimodal (video, audio, text) perception"***, what about Perceiver [*]?
> > > >
> > > > [*] Perceiver: General Perception with Iterative Attention. Jaegle et al.

---

> > > > > ### Author Response · Authors · 2021-09-02
> > > > > **Response to Reviewer GNEo**
> > > > >
> > > > > We appreciate the reviewer’s valuable points.
> > > > >
> > > > > ### Benefit of Transformer
> > > > > We do agree that the core benefit of using Transformer in our paper is the fact that Transformer is input-independent and one model could be shared across multiple modalities. We make sure to mention this in our manuscript.
> > > > > We would like to emphasize that one of the other benefits of Transformer especially compared to MLP-Mixer (that reviewer taBL had mentioned) is the fact that the scalability of Transformers and their generalizability of them are more explored and require less ad-hoc designs. We do consider studying the efficacy of MLP-Mixer on video, audio, and language understanding in our future work.
> > > > >
> > > > > Another benefit that we would like to mention is that Transformers are extremely efficient on advanced accelerators (even though they usually have more parameters compared to ConvNets). We observe a significantly less requirement for the number of TPU days for training VATT compared to its ConvNet baseline (MMV).
> > > > >
> > > > > ### Perceiver
> > > > > Thanks for mentioning Perceiver. The Perceiver (which is a concurrent work to ours) focuses on providing a framework for **unified raw input processing** with linear attention. Perceiver does **not** provide a pipeline for single-backbone multimodal perception and to the best of our knowledge, all experiments have been performed separately and the modalities do not share the same weights.
> > > > > What we are claiming is that we are the first to provide a unified model that does **share weights across multiple modalities**, while still operating on **raw inputs**. Such study has not been done in the literature.

---

### Official Review · Reviewer_GNEo · 2021-07-19

**Rating:** 6
**Confidence:** 4

**Summary:**

The paper proposes a transformer-based architecture for learning representations from video-audio-text triplet data without manual data annotation. The paper studies modality-specific transformers (one transformer for one modality) and modality-agnostic transformers (a shared transformer for all modalities). The learned representation is evaluated on various downstream tasks on image, video, audio and video-text.

**Ethical Concerns:**

No major ethical concerns.

**Limitations And Societal Impact:**

As stated above, one of the key contributions is a transformer-based multimodal architecture, however it does not outperform the ConvNet counterpart when using the same loss. Discussion on this point would be valuable.

The proposed modality-agnostic transformer is novel, and the paper shows its performance on downstream tasks. But it’s still not clear to me what can be captured in that unified transformer. I am curious to see more qualitative results like the attention maps per encoder layer for each modality.


**Main Review:**

The originality of the paper is limited. The standard VATT model is identical to [1] (Multimodal versatile network, Alayract et al.) except changing the ConvNet backbones into transformers. The novel point is the modality-agnostic shared transformer for all three modalities. The proposed DropToken is a useful technique to reduce the computation overhead for transformers.

The paper shows a large number of experimental results, especially for downstream tasks. The overall quality is good.

The paper is clearly written.

The significance of the results is moderate:
1. The key method of the paper is a transformer-based architecture for multimodal learning. In detail, it uses identical architecture and loss design (sec 3.3, 3.4) except replacing the backbone with transformers. But when comparing with [1] (Table 4 and supp Table 2), the proposed transformer backbone underperforms or gives similar results as its ConvNet counterpart, also the discussion is limited on this point.
2. The downstream performance of modality-agnostic transformers is not far from modality-specific transformers, which is encouraging for future works.
3. Table 5 shows the proposed DropToken is useful for reducing computation while keeping the downstream performance.


[1] Self-supervised multimodal versatile network, Alayrac et al.


**Time Spent Reviewing:**

5

---

> ### Author Response · Authors · 2021-08-10
> **Response to reviewer GNEo**
>
> ## Q: Comparing VATT to MMV
> We thank the reviewer for pointing this out. We will clarify the contributions of our work over MMV to improve the presentation of the work and to resolve the concerns. Specifically, we will clarify the following:
> ### VATT has major distinction from MMV
> 1. Pure Transformer architecture
> 1. Modality-agnostic backbone (sharing one set of weights across multiple modalities)
> 1. Operating on raw inputs (e.g. MMV relies on spectrograms or pre-extracted word embeddings)
>
> ### VATT achieves superior performance compared to MMV
> A direct comparison with MMV's original numbers is not fair due to the ~20% shrinkage of HowTo100M. Because of that, we re-trained MMV using their official code and achieved different results. To avoid confusion, we did not report the numbers in the paper. However, after receiving this criticism, we will add those numbers back to the paper for the readers' reference.
> Our re-implementation of MMV’s best setting achieves the following results. We can see that training on the same amount of data and using the same framework, VATT achieves much better results.
>
> |  Model      | UCF-101  | HMDB-51 |   ESC-50 | YouCook2 | MSR-VTT |
> | ------------- |:-------------:|:-------------:|:-------------:|:-------------:|:-------------:|
> | MMV        |    84.6      |    63.6      |    81.2      |    29.3      |    21.1      |
> | VATT        |  **89.6**  |   **65.2**  |  **84.7**  |  **45.5**  |  **29.7**  |
>
> ## Q: Further analyses on modality-agnostic setting
> We indeed performed further analyses. Among the most interesting results were the following which shows per-node-per-layer activation of the network w.r.t input modalities (please click on the link).
> We can see that the network is selective for text vs. high-redundant modalities (video/audio). Interestingly, it activates with all modalities when it comes to its later layers. This observation suggests that the network first tries to extract modality-specific features, skips them through certain layers, and later in the output layers it activates on all modalities to further process semantic information in higher level. We will add this analysis and include our observations and discussion on it.
>
>
> Figure: [Modality-Agnostic GeLU activations](https://i.ibb.co/DpRVkZt/Screen-Shot-2021-08-10-at-4-30-48-PM.png)
>
> ![](https://i.ibb.co/DpRVkZt/Screen-Shot-2021-08-10-at-4-30-48-PM.png)

---

> > ### Comment · Reviewer_GNEo · 2021-09-01
> > **Reply to authors' response**
> >
> > Thanks for the authors' response. I read all reviews and responses.
> >
> > The authors' response addressed some of my concerns. However,
> > 1. I think the experiment table comparing with MMV above is not very reliable (20% shrinkage of HowTo100M data leads to huge performance drop for MMV) and probably not a fair comparison (same as the reviewer PWef).
> > 2. I agree with the reviewer taBL that more discussion about the motivation of using the transformer architecture is required.
> >
> > Overall, I decided to keep my original rating. I think the paper is *marginally* above the threshold, but marginal.

---

### Decision · Program_Chairs · 2021-09-27

**Decision:**

Accept (Poster)

**Comment:**

The merits and weaknesses of this submission were discussed at length in back-and-forth conversations with the authors. The overall sense is that, although the technical novelty of the submission is limited, the paper provides a valuable large-scale empirical study on sharing a single transformer backbone architecture across different modalities. Several reviewers comment on the importance of sharing such results with the community. The ACs find that this is particularly relevant given that few labs are equipped with the resources to carry out such a study (thus here the ACs dissent from the opposite point made by some reviewers). The submission also introduces DropToken, a simple strategy to significantly reduce the computational cost of training large-capacity transformers. Based on these contributions, the ACs agree to accept this submission but encourage the authors to incorporate the feedback provided by reviewers, especially as it concerns the motivation for using transformers and the novelty claims.